# Reduced Antibiotic Resistance in the Rhizosphere of *Lupinus albus* in Mercury-Contaminated Soil Mediated by the Addition of PGPB

**DOI:** 10.3390/biology12060801

**Published:** 2023-05-31

**Authors:** Daniel González-Reguero, Marina Robas-Mora, Vanesa M. Fernández-Pastrana, Agustín Probanza-Lobo, Pedro Antonio Jiménez-Gómez

**Affiliations:** Department of Pharmaceutical Science and Health, San Pablo University, CEU Universities, Ctra. Boadilla del Monte Km 5.300, 28668 Boadilla del Monte, Spain; vanesa.fernandezpastrana@ceu.es (V.M.F.-P.);

**Keywords:** antibiotic, biorremediation, cenoantibiogram, heavy metal, mercury

## Abstract

**Simple Summary:**

Mercury pollution represents a serious environmental and health problem. Additionally, it may lead to the selection of bacterial mechanisms of antibiotic resistance. The use of bacteria capable of improving plant development can help plants to better adapt to contaminated environments, contribute to the decontamination of these sites and prevent antibiotic-resistant bacteria from affecting animal and human health. The present study proposes a way to evaluate the beneficial effect that some bacteria can have in mitigating the spread of antibiotic resistance in mercury-contaminated soils. In the experiments carried out, we observed how inoculated bacteria can reduce resistance to antibiotics in the soil, suggesting their potential for minimizing the dispersion of these mechanisms of antibiotic resistance.

**Abstract:**

The emergence of antibiotic resistance (AR) poses a threat to the “One Health” approach. Likewise, mercury (Hg) pollution is a serious environmental and public health problem. Its ability to biomagnify through trophic levels induces numerous pathologies in humans. As well, it is known that Hg-resistance genes and AR genes are co-selected. The use of plant-growth-promoting bacteria (PGPB) can improve plant adaptation, decontamination of toxic compounds and control of AR dispersal. The cenoantibiogram, a technique that allows estimating the minimum inhibitory concentration (MIC) of a microbial community, has been postulated as a tool to effectively evaluate the evolution of a soil. The present study uses the metagenomics of *16S rRNA* gene amplicons to understand the distribution of the microbial soil community prior to bacterial inoculation, and the cenoantibiogram technique to evaluate the ability of four PGPB and their consortia to minimize antibiotic resistance in the rhizosphere of *Lupinus albus* var. Orden Dorado grown in Hg-contaminated soils. Results showed that the addition of A1 strain (*Brevibacterium frigoritolerans*) and its consortia with A2, B1 and B2 strains reduced the edaphic community´s MIC against cephalosporins, ertapenem and tigecycline. The metagenomic study revealed that the high MIC of non-inoculated soils could be explained by the bacteria which belong to the detected taxa,. showing a high prevalence of Proteobacteria, Cyanobacteria and Actinobacteria.

## 1. Introduction

Mercury (Hg) is a metal with a high level of toxicity that severely affects ecosystems [1,2]. It is incorporated and biomagnified in the food chain and, consequently, affects human health even at very low concentrations [3]. One of the areas with the highest Hg pollution in the world is located in the mining district of Almadén (Ciudad Real, Spain), where levels of up to 8889 μg kg^−1^ are reached [4]. In order to give alternative uses to contaminated soils, the scientific community seeks to develop actions to mitigate the effects of Hg. Several physicochemical methods have been proposed, but the current trend is to use more sustainable methods, based on biotechnological techniques such as bioremediation. Therefore, there is a growing interest in the selection of microbial strains with potential bioremediation use [5]. Specifically, recent research has focused on the search for plant-growth-promoting (PGP) microorganisms, for their ability to promote plant growth and stimulate their bioremediation capabilities [5,6,7,8]. Recently, a great variety of studies have been reported in which methods for the selection of these microorganisms are proposed [6,9,10].

Likewise, it is known that soil microbial communities subjected to high abiotic pressure (such as heavy metal contamination) act as a natural reservoir of antibiotic resistance genes (ARGs) [11,12]. Several studies describe co-selection mechanisms of resistance to various toxic compounds and ARGs, especially co-resistance to heavy metals and antibiotics [10,12,13]. Antibiotic resistance (AR) is an emerging global problem that has attracted the attention of the scientific community in recent years. Its development and evolution in the clinical environment is evident [14,15]. Numerous genes that enable antibiotic resistance found in pathogenic bacteria have evolved or have been acquired from environmental microbial communities [16]; thus, the presence of multiresistant bacteria outside the hospital setting has been reported. This fact suggests the need to study and understand how the environment can behave as a reservoir of AR mechanisms. Soil provides habitat for many species that naturally produce substances with antimicrobial potential, such as *Bacillus* sp. or *Streptomyces* sp. Therefore, most AR mechanisms have an environmental origin. In nature, antibiotics can, at sub-inhibitory concentrations, exhibit different functions, such as the activation/deactivation of virulence factors or the regulation of microbial communication systems [17,18,19,20]. The positive selection of bacterial mutants in response to contaminants, such as Hg or antibiotics, could increase the mutation rate. In this way, antibiotics can act as an evolutionary force for the production and selection of new resistance mechanisms [19]. The presence of chromosomal genes coding for resistance mechanisms explains, in part, that many bacteria, even in natural antibiotic-free environments, can naturally carry a large number of ARGs [19,21]. 

These ARGs can be transmitted to bacteria with clinical relevance, and new multiresistant bacteria may appear [22,23,24,25] threatening the “One Health” approach. For this reason, the World Health Organization (WHO, Geneva, Switzerland) has declared ARGs to be a new pollutant due to their emerging prevalence and wide distribution. Goal 2 of the “Global Action Plan on Antimicrobial Resistance” also sets out the need to strengthen knowledge and the scientific base of AR through monitoring and research. It highlights the importance of increasing knowledge of the emergence and spread of antibiotic resistance among humans and animals through the environment. It also highlights the importance of developing new research tools aimed at expanding knowledge in agriculture and aquaculture, to combat the growing impact of antimicrobial resistance [26].

The expression of ARGs in different populations may be due to various factors, including the additive effect of different strains mediated by microbial communication processes such as quorum-sensing/quorum-quenching [20,27], ecological competition [28], as well as the response to abiotic factors, as in the case of the present work in which the influence of heavy metals is analyzed. Ecological competition occurs when individuals directly harm each other. In microorganisms, it refers to the secretion of metabolites that directly affect the proliferation of others, such as the secretion of antibiotics and asphyxiating polymers [28]. Co-culture experiments have shown that these secreted factors often determine which populations may prevail in complex communities [29], affecting microbial diversity [30]. Bacteria have developed methods that allow them to detect and respond directly to ecological competition by developing and selecting resistance mechanisms. In the response to biotic stress, bacteria interact with each other and regulate a set of behaviors favorable to their survival. Many of these responses, as well as their phenotypic expression, are regulated by well-known mechanisms of microbial communication like quorum sensing/quenching mechanisms [31]. 

In the same way that biological competence favors the emergence and selection of AR mechanisms, several studies demonstrate the contribution of metals in the co-selection of ARGs [10,13,32]. In a strongly polluted environment, competition processes between populations favor the selection of a greater number of AR mechanisms. Likewise, other pollutants, such as antibiotics, are constantly being released into the environment as a result of anthropogenic activity. This results in antibiotic pressure for the selection of resistant strains, favoring the mobility of ARGs [33,34,35,36,37]. Therefore, it is of special interest to discriminate, according to the different AR mechanisms, those antibiotics that can be used as biological indicators. 

Taking into account these factors, it is of special interest to search for: (i) tools that allow us to perform an analysis of the resistances of a microbial community and its potential behavior as a reservoir of resistance; (ii) microbial agents capable of mitigating the effects of contaminants on soil, as well as reducing the expression of antibiotic resistance in soils. 

One of the proposed tools is the so-called cenoantibiogram, which is defined as the phenotypic study of antibiotic resistance of a complete microbial community [38]. The evaluation and monitoring of microbial biocenosis requires tests for environmental microbiological control. This technique is postulated as a bioindicator of the evolution of the edaphic community, as well as the comparison of different communities. In this sense, the application of the cenoantibiogram concept opens the possibility of using a new tool to evaluate the effect of bioremediation treatments on complex bacterial communities. Likewise, the combination of this new tool with metagenomic techniques for *16S rRNA* analysis provides an excellent opportunity to evaluate changes in the composition, diversity and structure of the soil microbial population.

The present study aims to interpret and compare the impact of the use of four PGP bacteria (PGPB) and their respective consortia on the quality and microbiological diversity of the rhizosphere of *Lupinus albus* in soils with high concentration of Hg for further environmental bioremediation assays.

## 2. Materials and Methods

### 2.1. Study Area

Analyzed soils were taken from the mining district of Almadén (Ciudad Real, Spain). Specifically, the “S” slope of Cerro Buitrones in “Plot 6”, described by other authors in previous studies [39], was sampled. The concentration of Hg in this plot is 1710 mg kg ^−1^ total Hg, 0.609 mg kg ^−1^ soluble Hg and 7.3 mg kg ^−1^ interchangeable Hg. 

### 2.2. PGPB Isolation, Selection and Characterization

The strains used for this study were isolated from bulk soil and the rhizosphere of plants naturally grown in Plot 6 in the mining district of Almadén in Ciudad Real, Spain [39]. The strains were selected based on their Biomercuroremediation Suitability Index (BMRSI) values [9], which jointly evaluates PGPB activities and their tolerance to Hg. Hg tolerance was quantified by calculating the minimum bactericidal concentration (MBC). The PGP activities analyzed were: auxin production (3-indoleacetic acid: IAA) [40], presence of the enzyme 1-animocyclopropane-1carboxylate decarboxylase (ACCd) [41], siderophore production (SID) [42] and phosphate solubilizing capacity (PO_4_^3−^) [43]. The BMRSI was calculated according to the following formula, where 1 and 0 indicate presence or absence in the variables ACCd and PO_4_^3−^:BMRSI = [IAA (µg mL^−1^) + ACCd (1/0) + SID (cm) + PO_4_^3−^ (1/0)] + [MBC Hg (µg mL^−1^)]

The potential PGPB capacity in the presence of Hg of the four bacterial isolates (Table 1) was analyzed by González et al. [6]. For the present study, a BMRSI value > 6.5 was used as a selection criterion for the strains to be tested. For the trials, they were used individually, as well as the combination consortia in pairs, as provided in Table 2. 

The compatibility of the consortium strains was previously checked by means of the mutual compatibility test by the cross-streak method [44] in standard method agar plates (SMA, Pronadisa^®^, Madrid, Spain).

### 2.3. Biological Assays

Seeds of *Lupinus albus* var. Orden Dorado from the seed bank of the Center for Technological and Scientific Research of Extremadura were used.

As a preliminary step, the seeds were soaked in sterile distilled water and preserved at 4 °C for 24 h. The surface was sterilized with three washes of 70% ethanol for 30 s [45]. For pregermination, seeds were placed in trays with sterile vermiculite and irrigated with sterilized tap water until field capacity was reached. Under these conditions, they were kept in darkness for 72 h at 25 °C. Seeds with an emerged radicle of 3 cm ± 0.2 cm were selected for the study.

For the biological tests, sterile forest trays were used (Plásticos Solanas S.L., Zaragoza, Spain), each composed of 12 alveoli 18 cm high, with a capacity of 300 cm^3^, and a light of 5.3 cm × 5.3 cm. Eleven trays were used. To avoid cross-contamination, four pregerminated seeds were sown in each alveolus. A single bacterial strain (or consortium) and/or control was inoculated in each tray, so that 48 seeds were tested for each treatment. 

For bacterial treatment, a bacterial suspension was performed in 0.45% saline, and the inoculum density was adjusted to 0.5 McFarland. Each seed was inoculated with 1 mL of the suspension.

A plant growth chamber (phytotron) equipped with white and yellow light with photoperiod of 11 h of light, light intensity: 505 μmol m^−2^ s^−1^, temperature 25 ± 3 °C was used. Irrigation was carried out every 48 h by capillarity with sterile tap water, with an experimental volume of 350 mL/tray (12 alveoli).

After a growth period of 21 days, the plants were extracted from the inoculated soils by collecting the fraction of the soil intimately linked to the root. For each treatment, the rhizospheric soil was collected (1–2 g per plant) and homogenized to constitute the 60 g analysis sample. The homogenized sample was divided into three technical replicates. For the extraction of the microbiota from each sample, the procedure described by Velasco et al. [46] was modified. To do this, 2 g of soil of each of the technical replicates was suspended in 20 mL of sterile saline solution (NaCl 0.45%) and homogenized with an Omni-Mixer homogenizer at 16,000 r.p.m. for 2 min. It was then centrifuged at 690× *g* for 10 min with a Hettich Zentrifugen centrifuge model Mikro 22R. The supernatant was collected for the cenoantibiogram study. The remaining rhizospheric fraction of non-inoculated plants (approx. 60 g per treatment) and bulk soil (control soil) was separated into three technical replicates for metagenomic study.

### 2.4. DNA Isolation

The DNA was purified using a DNeasy Power Soil Pro Kit (Qiagen, Germantown, MD, USA) following the manufacturer’s instructions. An enzyme lysis step with lysozyme was included to obtain the highest and best amount of total bacterial DNA. Purified DNA was quantified using PicoGreenTM (ThermoFisher Scientific, Waltham, MA, USA) from 40 pg. DNA isolated from each sample was used for metagenomic analysis.

Two hypervariable regions (V3–V4) of the *16S rRNA* gene were amplified using primers (341F-5’CCTACGGRRBGCASCAGGKVRVGAAT; 785R-5’GGACTACNVGGTWTCTAATCC). 

### 2.5. Data Analysis and Bioinformatics

The purified amplicons were sequenced. Paired-end sequencing was done on an Illumina Mi-Seq platform. Bioinformatic analysis and quality control were performed using the Fast QC tool [47]. Raw sequence reads underwent quality trimming using Trimmomatic to remove adaptor contaminants and low-quality reads. DADA2 software was used to check for chimeras. The Q-score was used to predict the probability of an error in base-calling. Over 85% of bases >Q30 averaged across the entire run was considered acceptable. OTUs (operational taxonomic units) were identified from all reads using the QIIME2 software package, and a representative sequence for each OTU was also constructed.

The FastQ sequences were deposited in the BioProyect repository, under accession numbers PRJNA934906 for Cont_S and PRJNA934908 for Cont_P.

### 2.6. Cenoantibiogram: AR Profile of the Microbial Community

From the soil extract obtained in saline solution (NaCl 0.45%), it was verified that the density of viable microorganisms was >10^8^ ufc mL^−1^ (optical density [OD] = 0.5 McFarland). The bacterial extraction was sown in Mueller–Hinton agar (Condalab^®^, Madrid, Spain), and the minimum inhibitory concentration (MIC) was evaluated by the Kirby–Bauer method, using ε-test antibiotic strips, in triplicate, for the following antibiotics: cefuroxime, cefuroxime axetil, cefoxitin, cefotaxime, ceftazidime, cefepime, ertapenem, imipenem, amikacin, gentamicin, nalidixic acid, ciprofloxacin, tigecycline and trimethoprim/sulfamethoxazole (BioMérieux^®^, Marcy l’Etoile, France). Plates were then incubated according to the manufacturer’s instructions. For the quantification of the MIC, the most restrictive halo was used as reference.

### 2.7. Statistical Analysis

To evaluate the quality of the technical replicates in each soil, a Pearson correlation (r) of the percent genus abundances was done. The Kolmogorov–Smirnov test was performed to check the normality of all variables. Subsequently, a one-factor ANOVA and a post-hoc Kruskal–Wallis analysis were performed. Similarly, a principal component analysis (PCA) was performed, starting with the 2D projection of the load factors. All statistical differences refer to the comparison of the variables manifested by plants according to their inoculum against non-inoculated soil and plant controls. SPSS (Version 27.0 IBM Corp, Armonk, NY, USA) was used for all statistical analyses.

## 3. Results

### 3.1. Metagenomic Analysis

In order to evaluate the edaphic microbial diversity, prior to bacterial inoculation, a metagenomic analysis of amplicons of the *16S rRNA* gene was carried out to obtain the relative composition of the taxa that inhabit both the free soil (Cont S) and the rhizospheric soil (Cont P). In the metagenomic extraction of DNA and sequencing of the samples, between 98% and 97% of the sequences were maintained after the QC analysis (quality control). The abundance of species between technical replicates was highly correlated (all comparisons had an r > 0.99 with Pearson’s correlation test). Of the readings obtained in Cont S, 32.7% could not be assigned to any taxon. Likewise, 16.7% could not be assigned in Cont P.

A taxonomic diversity study was conducted using the Simpson (D) and Shannon (H’) diversity indices, which revealed that the diversity in rhizospheric soils is lower compared to free soil (Table 3).

The taxonomic analysis (Figure 1A, Appendix A) showed a greater representation of Proteobacteria and Actinobacteria, as well as Firmicutes, Planctinomyces, Acidobacteria and Cyanobacteria in Cont S versus Cont P. It is worth noting the low proportion of the taxonomic fraction corresponding to Firmicutes, mainly in Cont P. In contrast, Streptomycetales and Rhizobiales appear in greater proportion in Cont P (Figure 1B, Appendix A). Finally, the presence of sequences associated with viruses were similar in both samples.

### 3.2. Antibiogram

Before the release of any biological agent into the natural environment (PGPB), its biosafety must be guaranteed (both in handling and for the health of animals and plants). To this end, an antibiogram of PGPB strains was performed, which includes many of the most widespread therapeutic antibiotics (Table 4).

### 3.3. Cenoantibiogram

Cenoantibiograms of each of the treated soils were carried out. These results were then compared with the cenoantibiograms obtained from the soils without biological treatment. Table 5 shows that the soil microbial community naturally presents high MICs to the cephalosporin antibiotic group. Inoculation with the studied PGPB and their consortia results in a variation in the soil resistance profile. This effect is especially noticeable in the case of A1 strain and its respective consortia CS1, CS2 and CS3. The main affected antibiotics regardless of the treatment are ertapenem (carbapenemmic), and tigecillin (glycylcycline). A statistically significant reduction is observed for the antibiotic cefepime between Cont S compared to biological treatments with bacterial inoculation. 

To discriminate the overall behavior of the soils subjected to the different biological treatments, a statistical principal components analysis (PCA) was carried out. Figure 2 shows the 2D graph of the load factors. Soils treated with A1 strain (*Brevibacterium frigoritolerans)*, as well as its respective consortia CS1, CS2 and CS3, are segregated from the rest of the samples (which maintain a greater homology with the MICs of the controls). Table 6 shows that the accumulation of two factors explains the model with a cumulative variance greater than 95%.

Figure 3 shows the load factor of the PCA model ordered by antibiotics and projected in 2D. It can be observed that the data are separated into two large subsets on the abscissa, with positive values of this axis representing those antibiotics to which there is a higher MIC (A). In contrast, those antibiotics that register lower MIC values are in the negative values of this axis (B). It is interesting to note that all antibiotics whose resistance is associated with point mutations or enzymes of metabolism are segregated (negative abscissa axis) (C). On the other hand (negative abscissa axis), we find the group of cephalosporins, whose resistance in the environment is explained by microorganisms owning cephalosporins (D).

## 4. Discussion

There are several methods for the remediation of heavy-metal-contaminated soils (such as Hg). Currently, biological methods are considered more respectful towards the environment, especially those based on biotechnological techniques such as bioremediation. For this reason, there is a growing interest in the use of bioremediation, either through the use of plants, microorganisms or both (phytorhizoremediation) for the recovery of these environments [5,6,48,49,50]. There are multiple studies on the use of microorganisms for the recovery of heavy-metal-contaminated areas [5,8,48,51,52,53,54] and the benefits of adding PGPB that favor the process. Likewise, it is important to know the impact that their addition generates in the composition and biological diversity of the communities that host them, as well as the impact that the addition of microorganisms can exert on the expression of AR mechanisms of microbial communities. The use of massive sequencing techniques can help to better interpret these results. In the present study, the analysis of *16S rRNA* was carried out at the genus level, since, as several studies have pointed out [55,56], amplicon metagenomics do not have sufficient resolution to provide reliable information at the species level. This is due to the high heterogenicity of existing and yet to be described species. 

However, there is a consensus in affirming that the metagenomic analyses of the community allow a better integral interpretation of the bacterial composition of the edaphic ecosystems. 

There are metagenomic and functional analyses related to the soils tested in this work [57,58] in which a high representation of Actinobacteria and Proteobacteria is described, in both soils, but with a higher abundance in Cont P (Figure 1A, Appendix A). These studies also show the low representation of bacteria belonging to the phylum Firmicutes, particularly in Cont P (Figure 1). In contrast, we found a wide representation of the phylum Cyanobacteria in Cont S (Figure 1A, Appendix A), a fact described in previous tests in these soils. The presence of this phylum has been traditionally associated with aquatic environments but is a singularity in edaphic ecosystems. Its presence is probably related to greater resistance to Hg, as in other Gram-negative groups [59]. Similarly, we found an evident relevance of the genus *Pseudomonas* in these soils (Figure 1C, Appendix A). *Pseudomonas* have the ability to adapt and integrate into a wide variety of ecosystems, and their resistance to this type of medium contaminated with Hg, as well as their decontamination capacity, is well documented [60,61,62,63].

The results agree with the data provided by other authors who showed how the taxa of Gram-negative bacteria, and especially Alpha-proteobacteria, tend to have greater representation than Gram-positive ones in these environments influenced by the presence of Hg, precisely because of their high MIC compared to this contaminant (Figure 1) [57,58]. 

Although the genera *Bacillus* and *Clostridium* are considered as habitual members of edaphic communities, their abundance in the present study is very low or none, a fact already manifested in previous studies [57,58]. In contrast, we found a strong presence of *Streptomyces,* which indicates the colonizing capacity of some Gram-positive taxa.

As can be seen in Table 3, the incorporation of exogenous biological agents such as plant roots exerts a depressant effect on the soil microbiota in Cont P. Similarly, several authors have described how inoculation of a microorganism has a diversity-reducing effect [64,65,66,67]. These characteristic changes in the distribution and activity of the microorganisms associated with the root, have been called “rhizosphere effect” [68,69,70,71]. The composition of the rhizospheric community depends directly on the root exudates, as well as on the plant species, the type of root, the age of the plant, the phenological state, and the type and historical use of the soil.

The results regarding the behavior of microbial communities against antibiotics coincide with those described in previous studies. In these studies, mechanisms of RA that correspond to the phenotypic profile observed in the present study were detected, particularly in the MIC against carbapenems, cephalosporins and fluoroquinolones [57,58]. In the inoculated soils, there is a variation in the profile of the cenoantibiogram, which has as its most plausible cause a substitution in the composition and relative distribution of the original edaphic microbiota. In this same sense, other studies in which similar approaches are tested concluded that the taxonomic groups of inoculated microorganisms significantly increase their presence in the community that hosts them [67,72].

The four bacteria used in this study have a high bioremediation potential and have been proven effective as PGPB in Hg-contaminated soils (Table 1) [6,48]. The PGPB capacity of *Brevibacterium frigoritolerans* strains (A1 in this study) is well described [73,74]. In some studies, strains resistant to various antibiotics have been described and may even have ARGs of high clinical relevance, such as extended-spectrum β-lactamase-, cefotaxime- and vancomycin-resistance genes [75,76]. On the contrary, the A1 strain had a low MIC profile to most of the antibiotics tested. Although there are few references that link it as a potential infectious agent [76,77], it could act as an ARG transmission agent up to clinically relevant strains. Some strains of *Bacillus toyonensis* manifest ARs of clinical relevance, such as cefotaxime, trimethoprim, ampicillin and various β-lactamases [78,79]. In contrast, the A2 profile (tested in the present study) presents low MIC to carbapenems, aminoglycosides and fluoroquinolones (Table 4). Both *Pseudomonas mercuritolerans* (strain B1) and *Pseudomonas baetica* (strain B2) are very poorly known strains due to their recent description [61,80], such that in the bibliographies consulted, there is no information about their AR profile. Similarly, both strains have a sensitivity profile to carbapenems, aminoglycosides and fluoroquinolones, although they could be carriers of cephalosporinases. 

The absence of methods that allow global analyses of antibiotic resistance has not yet allowed a precise analysis of the impact that the addition of microorganism produces in the values of MIC compared to different antibiotics in the microbial communities that host them. The term ‘cenoantibiogram’ refers to the phenotypic profile of resistance to different antibiotics in a community, that is, to the behavior of the population as a whole. Thus, knowing the cenoantibiogram of an edaphic community contributes to a more detailed knowledge of its phenotypic behavior subjected to different factors of change [38]. 

The addition of the A1 strain (*Brevibacterium frigoritoleras*) has the ability to significantly reduce the MIC of the bacterial community of the soils against all antibiotics tested except quinolones (nalidixic acid and ciprofloxacin) (Table 7). Likewise, soils inoculated with any of the four strains show a significant reduction in their MIC compared to ertapenem (carbapenem beta-lactam) (Table 7). These results show the ability of the PGPB tested to oppose biological processes such as those described by other authors [10,11,12,13] who claim that the genes that provide resistance to Hg and those that provide AR are co-selected, conferring antibiotic–metal co-resistance.

The same results were found in the significant reduction in MIC values compared to tigecycline (glicilcycline) of the edaphic bacterial communities when any of the strains tested were added (Table 7). These results demonstrate the ability of tested PGPBs to reverse observations such as those of Rasmussen and Sørensen [81], who found that high levels of AR to tetracycline in environments with high Hg contamination could also be due to the transfer of conjugative plasmids. However, the experiment in the present work does not contradict what is reported in the literature. The phenomenon of reduced antibiotic resistance in a community can be explained by the displacement that the inoculated strains exert on the rest of the edaphic microorganisms, inducing functional changes that are evidenced by this decrease in MIC of the community. 

Conversely, the contribution of antibiotic-resistant bacteria to the environment can induce higher MICs. This is evidenced in the results of a cenoantibiogram of the analysis of different styles of cultivation and fertilization of *Vitis vinifera*, showing that those soils with a greater intervention (fertilized with fertilizer of animal origin or soils near farms) express higher MICs and greater resistance to antibiotics [38]. We therefore consider the convenience of verifying the profile of antibiotic resistance of a PGPB prior to considering its use in environmental recovery processes or agricultural or forestry exploitation. The use of a cenoantibiogram can contribute to a better understanding of the behavior of soil, whose phenotype can be the result of different causal factors:i.Heterogeneous composition in terms of species and strains of the community in a phenomenon of cooperative inactivation [82,83,84].ii.Variability in the numerical quotas of each of the populations that make up the community [85,86,87].iii.Processes of competition and/or intra- or interspecific synergy that modulate the expression of the ARGs presented by the populations [20,27].iv.Competition between populations for environmental resources [27,31].v.Interaction of abiotic components with the microbial community (soil environmental conditions, pH, moisture or salinity) [88,89,90,91].vi.Interactions, synergistic or antagonistic, between biomolecules that serve as mechanisms of resistance of the strains [17,19].

Both the data provided by the ANOVA of a Kruskal–Wallis (Table 7) factor and the PCA show how the A1 strain and its consortia have the ability to significantly modify the cenoantibiogram of the soils. This result suggests that this strain as a very good candidate to alleviate the stress caused by antibiotics that a soil may suffer, with the consequent effect of alleviating the possible spread of resistance to antibiotics for therapeutic use. Similarly, the CS3 consortium, made up of strains A1 and B2, has shown in previous studies [48] its PGP capacity in *L. albus* var. Golden Order in soils highly contaminated with Hg. This bacterial consortium is able to stimulate plant growth by improving the total weight of the plant, root weight, number of roots and number of leaves. This fact, together with the results of the present study, postulate it as a good candidate for use in the bioremediation of Hg as a promoter of the reduction of MIC values. 

## 5. Conclusions

The metagenomic analysis of soils with high concentrations of Hg shows a relative proportion of taxa belonging to Gram-negative bacteria, especially belonging to the Proteobacteria and Cyanobacteria groups, and Gram-positive Actinobacteria. This fact could justify the high MIC found in soils compared to the bacteria belonging to these taxa.

The addition of the A1 strain (*Brevibacterium frigoritolerans*) in isolation and as a consortium (CS1, CS2 and CS3) reduces the MICs to antibiotics of the edaphic community in studies with soils contaminated with Hg. The main antibiotic groups whose MIC is significantly reduced are cephalosporins, ertapenem and tigecycline.

The CS3 consortium (*Brevibacterium frigoritoleras* + *Pseudomonas baetica*), which was successful in promoting plant growth in soils contaminated with Hg in previous trials, is postulated to have high bioremediation potential by significantly decreasing the values of MICs of the community that hosts them against the antibiotics cefuroxime, cefotaxime, ertapenem, ciprofloxacin and tigecycline.

The results obtained in this study open a new horizon in the study of microbial communities through the study of the phenotypic profile of antibiotic resistance. In the same way, new paths are also opened for the study of the biosecurity of releasing microorganisms into the environment both in bioremediation processes and in the promotion of plant growth.

## Figures and Tables

**Figure 1 biology-12-00801-f001:**
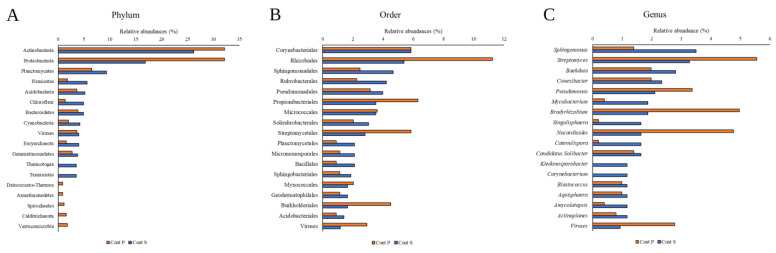
Relative abundances of the most representative taxa in the metagenomic analysis of amplicons of Cont S and Cont P, corresponding to the taxonomic grouping of Phylum (**A**), Order (**B**) and Genus (**C**). Relative abundance data are available in the Appendix A (Appendix A: Phylum; Appendix A: Order, and Appendix A: Genus).

**Figure 2 biology-12-00801-f002:**
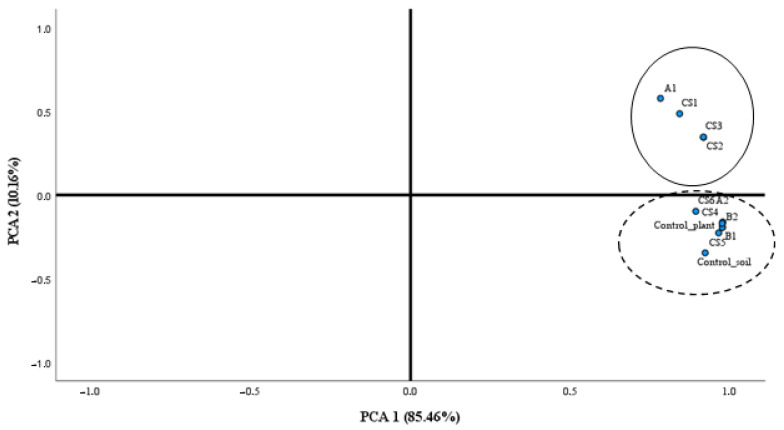
Representation of strains, consortia and controls according to PCA load factors (PCA1 85.46% and PCA2 10.16%) represented in Figure 3.

**Figure 3 biology-12-00801-f003:**
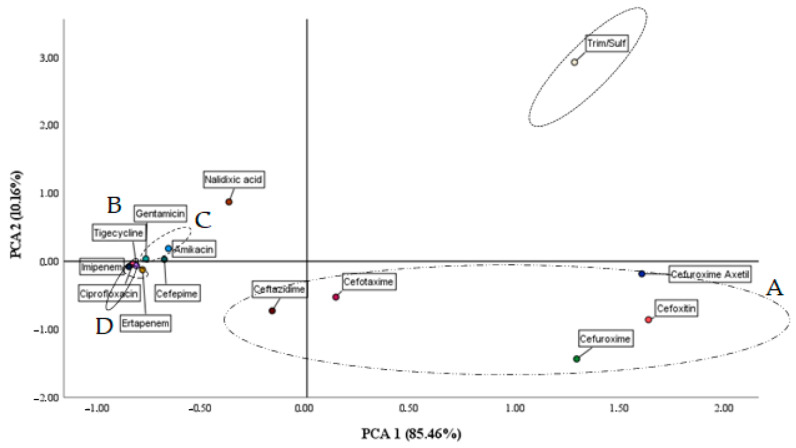
2D projection of PCA load factors and the MICs of the antibiotics studied.

**Table 1 biology-12-00801-t001:** Bacterial isolates according to their BMRSI in the presence of Hg [6].

Strain	HgCl_2_ Tolerance (µg mL^−1^)	BMRSI	Strain Origin	*16S rRNA* Identification
A1	140	6.54	*Avena sativa*	*Brevibacterium frigoritolerans*
A2	140	7.30	Bulk soil	*Bacillus toyonensis*
B1	140	7.20	Bulk soil	*Pseudomonas mercuritolerans*
B2	140	6.92	*Avena sativa*	*Pseudomonas baetica*

**Table 2 biology-12-00801-t002:** Consortia formed by the combination of the PGPBs included in the Table 1.

	CS1	CS2	CS3	CS4	CS5	CS6
Strains	A1+B1	A1+A2	A1+B2	B1+A2	B1+B2	A2+B2

**Table 3 biology-12-00801-t003:** Comparative table of the diversity indices of Simpson (D) and Shannon (H’) for the samples studied. Cont S: Free control soil without plants; Cont P: Rhizospheric control soil without inoculation.

Soil	D	H’
Cont S	0.985	4.464
Cont P	0.982	3.998

**Table 4 biology-12-00801-t004:** MIC (μg mL^−1^) of each of the four PGPB strains under study.

	A1	A2	B1	B2
*16S rRNA* identification	*Brevibacterium frigoritolerans*	*Bacillus toyonensis*	*Pseudomonas mercuritolerans*	*Pseudomonas baetica*
Cefuroxime	1	16	16	16
Cefuroxim eAxetil	1	8	16	16
Cefoxitin	8	8	16	16
Cefotaxime	1	8	8	8
Ceftacidime	0.5	8	4	4
Cefepime	1	1	2	2
Ertapenem	0.5	0.5	0.5	0.5
Imipenem	0.25	0.25	0.25	0.25
Amikacin	2	2	2	2
Gentamicin	1	1	1	1
Nalidixic Acid	4	4	2	2
Ciprofloxacin	0.5	0.5	0.25	0.5
Tigecyclina	0.5	0.5	0.5	0.5
Trimethoprim/Sulfomethoxazole	20	20	20	20

**Table 5 biology-12-00801-t005:** MIC (μg mL^−1^) of the different antibiotics studied from soils inoculated with strains and/or consortia and control soils without biological treatment.

Treatment	Cont S	Cont P	A1	A2	B1	B2	CS1	CS2	CS3	CS4	CS5	CS6
Cefuroxime	64	32	8	32	32	32	8	8	8	32	32	16
Cefuroxime Axetil	64	32	8	32	32	32	16	16	16	32	32	16
Cefoxitin	64	32	8	32	32	32	8	16	16	32	32	32
Cefotaxime	16	16	1	8	16	16	2	8	8	8	16	16
Ceftacidime	16	8	0.5	8	8	8	0.5	4	4	8	16	16
Cefepime	4	2	1	2	2	2	2	1	2	2	2	2
Ertapenem	4	4	0.5	0.5	0.5	0.5	0.5	0.5	0.5	0.5	0.5	0.5
Imipenem	0.25	0.25	0.25	0.25	0.25	0.25	0.25	0.25	0.25	0.25	0.25	0.25
Amicacin	2	2	2	2	2	2	2	2	2	2	2	2
Gentamicin	1	1	1	1	1	1	1	1	1	1	1	1
Nalidixic Acid	8	8	4	4	2	2	4	8	8	2	2	2
Ciprofloxacin	0.5	0.5	0.5	0.5	0.25	0.5	0.5	0.5	0.25	0.25	0.25	0.5
Tigecyclina	2	1	0.5	0.5	0.5	0.5	0.5	0.5	0.5	0.5	0.5	0.5
Trimethoprime/Sulfomethoxazole	20	20	20	20	20	20	20	20	20	20	20	20

**Table 6 biology-12-00801-t006:** The two main components that describe the model.

Component	Total	% SD	% Accumulated
1	10.26	85.46	85.46
2	1.22	10.16	95.64

**Table 7 biology-12-00801-t007:** ANOVA of a Kruskal–Wallis factor in which the MIC profile (μg mL^−1^) of the soil is compared after being inoculated with the different bacterial strains and/or their consortia, against Cont S Cont P.

Treatment	Cont S	Cont P	A1	A2	B1	B2	CS1	CS2	CS3	CS4	CS5	CS6
Cefuroxime	64	32	8 ^a,b^	32	32	32	8 ^a,b^	8 ^a,b^	8 ^a^	32	32	16 ^a^
Cefuroxime Axetil	64	32	8 ^a,b^	32	32	32	16 ^a,b^	16 ^a,b^	16 ^a,b^	32	32	16 ^a,b^
Cefoxitin	64	32	8 ^a,b^	32	32	32	8 ^a,b^	16 ^a^	16 ^a^	32	32	32
Cefotaxime	16	16	1 ^a,b^	8 ^a,b^	16	16	2 ^a,b^	8 ^a,b^	8 ^a,b^	8 ^a,b^	16	16
Ceftazidime	16	8	1 ^a,b^	8	8	8	1 ^a,b^	4 ^a^	4 ^a^	8	16	16
Cefepime	4	2	1 ^a,b^	2 ^a^	2 ^a^	2 ^a^	2 ^a^	1 ^a,b^	2 ^a^	2 ^a^	2 ^a^	2 ^a^
Ertapenem	4	4	0.5 ^a,b^	0.5 ^a,b^	0.5 ^a,b^	0.5 ^a,b^	0.5 ^a,b^	0.5 ^a,b^	0.5 ^a,b^	0.5 ^a,b^	0.5 ^a,b^	0.5 ^a,b^
Nalidixic Acid	8	8	4	4	2 ^a,b^	2 ^a,b^	4	8	8	2 ^a,b^	2 ^a,b^	2 ^a,b^
Ciprofloxacin	0.5	0.5	0.5	0.5	0.25 ^a,b^	0.5	0.5	0.5	0.25 ^a,b^	0.25 ^a,b^	0.25 ^a,b^	0.5
Tigecycline	2	1	0.5 ^a,b^	0.5 ^a,b^	0.5 ^a,b^	0.5 ^a,b^	0.5 ^a,b^	0.5 ^a,b^	0.5 ^a,b^	0.5 ^a,b^	0.5 ^a,b^	0.5 ^a,b^

“a” means a significant reduction (*p* ≤ 0.05) of antibiotic concentration versus Cont S; “b” means a significant reduction (*p* ≤ 0.05) of antibiotic concentration versus Cont P. Dark grey: Significant differences (*p* ≤ 0.05) with Cont S and Cont P; Light gray: Significant differences (*p* ≤ 0.05) with Cont P; and White: no significant differences (*p* ≥ 0.05).

## Data Availability

The data presented in the study are deposited in the BioProyect repository, under accession numbers PRJNA934906 for Cont_S and PRJNA934908 for Cont_P. Also see Appendix A.

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
