# Peer review of "Reduced Antibiotic Resistance in the Rhizosphere of Lupinus albus in Mercury-Contaminated Soil Mediated by the Addition of PGPB"

_biology, 2023, doi:10.3390/biology12060801_

Round 1
Reviewer 1 Report
The manuscript ‘Combined use of PGPB bacteria and Lupinus albus seedlings to minimize the antibiotic resistance profile in mercury contaminated soils, described interesting data. However, there are following points that needs elaboration.
L-185-187: rewrite
Cont S- sample is taken at which depth?
At many points abbreviations are explained, kindly check carefully.
Raw data files/accessions numbers not provided for sequencing and metagenomics analyses.
Plate assays pictures can be provided in supplementary data.
Did authors checked Hg concentration after inoculation, how much bio remediation was done in combination with plant and alone by bacteria? As well as correlation with AR profile.
It is mentioned that reported strains have PGPB ability, but plant growth promotion data not provided. Only bacterial biochemical assays are listed.
Author Response
Dear reviewer,
According to your review, you can find the revision in the file attached.

Reviewer 2 Report
The present paper presents a study on the use of PGPB bacteria and Lupinus albus seedling to reduce the antibiotic resistance in soils contaminated with mercury.
The main objective of this article is to show how addition of PGPR bacteria or consortium in a polluted soil modify the microbial diversity and how it alters the antibiotic resistance of the edaphic community.
The manuscript is accepted with minor revision
Introduction
Line 42, 63, 79: revision mark remains.
Line 43: Specify the location of Almedén.
Line 44, 133: use “kg” instead of “Kg”
Lines 124 to 127: the aim of the study seems to have no correspondence with the title. More specifically, the objective of the study as mentioned in the end of the introduction is limited to the study of the quality and microbiological biodiversity of the rhizosphere for further bioremediation tests.
Material and methods
Line 146: use PO43- instead of PO4-3 in the formulae
Conclusion
A few perspectives should be added to show how results open up new fields of research.
Author Response

(The authors gave the same response as above.)

Reviewer 3 Report
The work is interesting and can be accepted for publication after correcting some oversights.
The article entitled as “Reduction of antibiotic resistance in the rhizosphere of Lupinus albus in mercury contaminated soil mediated by the addition of PGPB”.
Antibiotic resistance of plant growth-promoting bacteria from Hg contaminated soils was studied in the presented work.
The work is interesting and can be accepted for publication after correcting some oversights.
1. First of all, resistance to antibiotics was studied in the work. In the work there was only a statement of the fact (without comparison) that the microorganisms were taken from mercury-containing soils. Therefore, both the introduction and summary should begin with the authors' research on antibiotics. And then point to the role of mercury.
The authors write (line 393): “the genes that provide resistance to Hg and those that provide AR are co-selected”. This is a good explanation for the abstract.
2. Line 140: The authors explain what the abbreviation MBC is. However, MIC (the first mention in the text is line 210) is not deciphered in the text; it is only in the abstract.
3. Lines 300-301: It is not clear what the authors mean when they write: “the possession of cephalosporins by microorganisms”.
4. Line 357: add white space into “fluoroquinolones[59,60]”.
5. Line 357: “ belonging to Gram-negative bacteria, especially belonging to the Actinobacteria,”. Actinobacteria are Gram-positive bacteria.
6. Lines 442-444: “The addition of the A1 strain (Brevibacterium frigoritolerans) both isolated and consortium (CS1, CS2 and CS3) reduces the MICs to antibiotics of the edaphic community of soils contaminated with Hg.” The phrase sounds like that in the work a comparison was made of soils contaminated with mercury with soils free of mercury. It would be more correct if instead of the words "of soils contaminated with Hg" a clarification would be made: " in studies with soils contaminated with mercury ".
The authors of the manuscript investigate the resistance to antibiotics of microorganisms that grew on a medium contaminated with mercury. The authors did not aim to compare with soils not contaminated with mercury.
Therefore, control measurements (this is from the comments of another
reviewer), in my opinion, are not required.
And in order not to mislead readers who are looking for comparison, I suggested restructuring the abstract and introduction. Perhaps the word "Reduction of" from the name should be removed and the "Antibiotic
resistance in the rhizosphere of Lupinus albus in mercury contaminated
soil mediated by the addition of PGPB" left.
I wish you success.
Author Response
The work is interesting and can be accepted for publication after correcting some oversights.
The article entitled as “Reduction of antibiotic resistance in the rhizosphere of Lupinus albus in mercury contaminated soil mediated by the addition of PGPB”.
Antibiotic resistance of plant growth-promoting bacteria from Hg contaminated soils was studied in the presented work.
The work is interesting and can be accepted for publication after correcting some oversights.
- First of all, resistance to antibiotics was studied in the work. In the work there was only a statement of the fact (without comparison) that the microorganisms were taken from mercury-containing soils. Therefore, both the introduction and summary should begin with the authors' research on antibiotics. And then point to the role of mercury.
The authors write (line 393): “the genes that provide resistance to Hg and those that provide AR are co-selected”. This is a good explanation for the abstract.
Abstract restructured
- Line 140: The authors explain what the abbreviation MBC is.However, MIC (the first mention in the text is line 210) is not deciphered in the text; it is only in the abstract.
Added on lines 210 - 211: “the minimum inhibitory concentration (MIC)”
- Lines 300-301: It is not clear what the authors mean when they write: “the possession of cephalosporins by microorganisms”.
Corrected on line 300 – 301: “the owning of cephalosporins by microorganisms”
- Line 357: add white space into “fluoroquinolones[59,60]”.
Done.
- Line 357: “belonging to Gram-negative bacteria, especially belonging to the Actinobacteria,”. Actinobacteria are Gram-positive bacteria.
Your appreciation is totally correct. That an error on the idea’s expression. Corrected on line 440: “the Proteobacteria and Cyanobacteria groups, and the Gram-postive Actinobacteria”
- Lines 442-444: “The addition of the A1 strain (Brevibacterium frigoritolerans) both isolated and consortium (CS1, CS2 and CS3) reduces the MICs to antibiotics of the edaphic community of soils contaminated with Hg.” The phrase sounds like that in the work a comparison was made of soils contaminated with mercury with soils free of mercury. It would be more correct if instead of the words "of soils contaminated with Hg" a clarification would be made: " in studies with soils contaminated with mercury ".
Phrase rewrite according to your contribution.
The authors of the manuscript investigate the resistance to antibiotics of microorganisms that grew on a medium contaminated with mercury. The authors did not aim to compare with soils not contaminated with mercury. Therefore, control measurements (this is from the comments of another reviewer), in my opinion, are not required. And in order not to mislead readers who are looking for comparison, I suggested restructuring the abstract and introduction. Perhaps the word "Reduction of" from the name should be removed and the "Antibiotic resistance in the rhizosphere of Lupinus albus in mercury contaminated soil mediated by the addition of PGPB" left.
We agree with that point, but one of the aims of the study is to evaluate the capacity to modulate the antibiotic resistance profile of the soil microbial community in order to use these bacteria in the Hg decontamination processes, reducing at the same time the potential reservoir antibiotic resistance of this soil.
I wish you success.
Reviewer 4 Report
Reduction of antibiotic resistance in the rhizosphere of Lupinus albus in mercury contaminated soil mediated by the addition of PGPB
Daniel González-Reguero1, Marina Robas-Mora1, Vanesa M. Fernández-Pastrana Agustín Probanza and Pedro A. Jiménez
General comments.
The authors present a study that attempts to determine which of 4 soil isolates with plant growth promoting properties can reduce the accumulation of antibiotic resistant bacteria in the rhizospheric soil of Lupinus albus grown on mercury contaminated soils. This addresses problems with metal contaminated soils serving as a stimulus for the spread of antibiotic resistance genes within the microbial community. If one could identify plant growth enhancing bacteria that also reduced antibiotic resistance in the rhizosphere then contaminant mitigation might be safer. In addition to the 4 isolates the authors also test all possible two-strain combinations of the 4 isolates. The protocol, as I understand it, involves planting seedlings of L. albus into growth trays inoculated with one of eleven treatments (one of the 4 isolates or one of the 6 two-strain consortia, + control with no treatment) followed by incubation and growth of plants. After 21 days the plants were uprooted and the rhizosphere soil extracted for metagenomic analysis and determination of antibiotic resistance to 14 antibiotics. In my reading of the M&M there should have been at least 11 metagenomic analyses but when one gets to the results there are only community analyses with 16S rRNA amplicons of free soil and rhizospheric soil. The authors report on these two metagenomes then report on the antibiotic resistance spectrum detected from the rhizospheric soils from the 11 different treatments.
First, the manuscript should be proofed for standard english. There were a number of awkward phrasing, problems with singular/plural and words inappropriately used. Second, the M&M needs to be carefully revised so that the reader can more easily understand the experimental design. Perhaps a figure or flow chart would assist. There are several aspects of the M&M that are not fully described. These include how the growth trays were set up (how much soil, spacing of the plants etc), how the soil was removed from the rhizosphere, the specifics of the metagenome analyses including how many sequences were generated and whether or not the samples were rarefied and what database was used for alignment and taxonomic ID. In addition it was not abundantly clear where the statistics were applied. Several figures where I thought there was replication lacked error bars or information regarding stats. There was apparently no biological replication for the treatments as the rhizopshere from 48 plants were "gathered and homogenized". This was followed by the construction of 3 technical reps. Note here that far more important than technical reps would be the biological reps. Third, the soil that was used was from a site that had previously been characterized and the mercury concentration determined. As far as I could tell, the new soil collected from the site for this study was not tested for [Hg]. This is infortunate because metals in contaminated soils can be very heterogenously distributed. It was also not clear what purpose the metagenomic studies served. These, as reported, were only performed on bulk and rhizospheric soils and it is not clear what they contribute to the punch line of the study other than a picture of what the community MIGHT look like with treatment. Metagenomic studies on the treatments would have been quite revealing given the results obtained from the antibiotic resistance studies, the PCA plot of which shows any rhizosphere treated with isolate A1 being distinctly different. The figure legends were rather sparse and did not provide the reader with sufficient information regarding the plotted data. In particular note the legend for Fig 2 which says "Figure 2. Representation of strains, consortia and controls according to PCA load factors (PCA1 290 85.46% and PCA2 10.16%)." PCA load factors of what? Table 7 probably isn't needed - just state the numbers in the text. While PCAs are great for a visual presentation, some stats to support it would be welcome. This is satisfied by their ANOVA analysis of the MIC profiles but some of the antibiotics/isolates had little statistical support and this is not shown on the PCA.
Overall these are challenging experiments and the focus of the study is sound and worthy of pursuit. However before publication there are a number of issues that require clarification.
Specific comments.
L2-3 As a general rule I would stay away from unexplained acronyms in the title.
L23-24 Recast this sentence.
L33-34 needs revision
L51 This is a bit awkward. I would delete 'both'.
L67 I agree. Still it is somewhat speculative.
L90-92 Ecological competition does not necessarily imply the secretion of growth inhibitors. It could occur simply as a consequence of competition for organic carbon.
L133 Were the new soil samples tested for [Hg]? Metal concentrations in soils can be very heterogenous. If you assume a concentration based on previous measures the concentration should be reconfirmed on your samples.
L182-184 By combining all of these samles you have reduced your replication to just three technical reps.
L152 consortia ?
L168. How much soil was used in each tray?
L185 This is not clear. Should be recast.
L204-205 Were the sequences checked for chimeras? What database and version were used for alignments and taxonomic ID? How many OTUs were identified for each sample? How many sequences for each sample? were the samples rarefied?
L187-188 This is incomplete. What happens after centrifugation? Which fraction is then of interest? Supernatant? Pellet? Complete the M&M.
L228-233 Repetition.
L227 This is not clear. In the M&M it sounds as though you have metagenomic samples from 10 treatments, not counting the soil control. Yet in this section you present only a soil control and soil rhizosphere. It also inidicates in the M&M that there are 3 technical reps but there appears to be no evidence of this in the results.
L210 How was it sown? How much was plated? What do the stats look like on the replicates strips?
L240-241 What did the three replicates look like. Not much difference in the Simpson. To prove to the reader that there is a real difference here you need stats on replicates. One can easily see 2-3 fold differences in OTU size just through the variability of PCR amplification.
Fig 1. This figure needs error bars.
L263 There is no indication of replication in this experiment. Need standard deviation or comparable. How many reps were there? How many technical reps and how many biological reps?
L286 Who? Are the samples a Who? Perhaps which would be better.
Fig 2 legend. More information is needed in the legends. What data set is this PCA based on?)
L317 What is this gender? Do you mean genus?
L334-335 Somewhat awkwardly expressed
L397 This seems like a large extrapolation.
L442 Needs to be rephrased.
Author Response
First of all, thank for your time reviewing our manuscript and for all your contributions in order to improve our work.
General comments.
The authors present a study that attempts to determine which of 4 soil isolates with plant growth promoting properties can reduce the accumulation of antibiotic resistant bacteria in the rhizospheric soil of Lupinus albus grown on mercury contaminated soils. This addresses problems with metal contaminated soils serving as a stimulus for the spread of antibiotic resistance genes within the microbial community. If one could identify plant growth enhancing bacteria that also reduced antibiotic resistance in the rhizosphere then contaminant mitigation might be safer. In addition to the 4 isolates the authors also test all possible two-strain combinations of the 4 isolates. The protocol, as I understand it, involves planting seedlings of L. albus into growth trays inoculated with one of eleven treatments (one of the 4 isolates or one of the 6 two-strain consortia, + control with no treatment) followed by incubation and growth of plants. After 21 days the plants were uprooted and the rhizosphere soil extracted for metagenomic analysis and determination of antibiotic resistance to 14 antibiotics. In my reading of the M&M there should have been at least 11 metagenomic analyses but when one gets to the results there are only community analyses with 16S rRNA amplicons of free soil and rhizospheric soil. The authors report on these two metagenomes then report on the antibiotic resistance spectrum detected from the rhizospheric soils from the 11 different treatments.
First, the manuscript should be proofed for standard English. There were a number of awkward phrasing, problems with singular/plural and words inappropriately used. Second, the M&M needs to be carefully revised so that the reader can more easily understand the experimental design. Perhaps a figure or flow chart would assist.
There was a graphical abstract, we don’t know why the last version on the mdpi portal doesn’t have it. Added for better understanding.
There are several aspects of the M&M that are not fully described.
These include how the growth trays were set up (how much soil, spacing of the plants etc),
The trays was conforming by 12 alveoli distributed in 6x2 rectangle. The four seeds contained in each alveoli was sow at 1.5cm (approximately) from the others in the same alveoli. Each tray was moved every 24 hours to ensure that all the plants receive the same light intensity and shadows during the growth phase.
how the soil was removed from the rhizosphere,
First the plant was removed from the tray and the excess of soil were discharged. The rhizospheric soil was collected by gently shaking of the plant in a sterile container.
the specifics of the metagenome analyses including how many sequences were generated and whether or not the samples were rarefied and what database was used for alignment and taxonomic ID. In addition it was not abundantly clear where the statistics were applied. Several figures where I thought there was replication lacked error bars or information regarding stats.
There was apparently no biological replication for the treatments as the rhizopshere from 48 plants were "gathered and homogenized". This was followed by the construction of 3 technical reps. Note here that far more important than technical reps would be the biological reps.
As it was explaining on the manuscript (lines 179 – 182), and as we have explained above, the experiment was carried out in a phytotron (which have the lights on one side and a reflection panel on the other) so the heat and the light that received each treatment it’s not the same. As well, the irrigation was done by capillarity, what means that not all the alveoli take water at the same rate during the experiment. These factors conduce us to make technical replicas instead of biological replicas, to study the general behavior of the strains throughout each treatment.
Third, the soil that was used was from a site that had previously been characterized and the mercury concentration determined. As far as I could tell, the new soil collected from the site for this study was not tested for [Hg]. This is infortunate because metals in contaminated soils can be very heterogenously distributed.
It was also not clear what purpose the metagenomic studies served. These, as reported, were only performed on bulk and rhizospheric soils and it is not clear what they contribute to the punch line of the study other than a picture of what the community MIGHT look like with treatment. Metagenomic studies on the treatments would have been quite revealing given the results obtained from the antibiotic resistance studies, the PCA plot of which shows any rhizosphere treated with isolate A1 being distinctly different.
The figure legends were rather sparse and did not provide the reader with sufficient information regarding the plotted data. In particular note the legend for Fig 2 which says "Figure 2. Representation of strains, consortia and controls according to PCA load factors (PCA1 290 85.46% and PCA2 10.16%)." PCA load factors of what? Table 7 probably isn't needed - just state the numbers in the text. While PCAs are great for a visual presentation, some stats to support it would be welcome. This is satisfied by their ANOVA analysis of the MIC profiles but some of the antibiotics/isolates had little statistical support and this is not shown on the PCA.
Corrected on the figures. Figure 3 correspond with the load factors of the PCA which explain the distribution of the strains and consortium.
Overall, these are challenging experiments and the focus of the study is sound and worthy of pursuit. However, before publication there are a number of issues that require clarification.
Specific comments.
L2-3 As a general rule I would stay away from unexplained acronyms in the title.
We totally agree with that appreciation, but we consider as well that “PGPB” is a common acronym in the microbiology study.
L23-24 Recast this sentence.
Phrase corrected to: The use of plant growth-promoting bacteria (PGPB) can improve plant adaptation, decontamina-tion of toxic compounds and control of AR dispersal
L33-34 needs revision
Sorry for the mistake, this information belongs to a previous version of the manuscript. Corrected to: “The metagenomic study revealed that the high MIC of non-inoculated soils could be explain by the bacteria which belong to the taxons detected. Showing a high prevalence of Proteobacteria, Cyanobacteria and Actinobacteria.”
L51 This is a bit awkward. I would delete 'both'.
Corrected by deleting “both”
L67 I agree. Still it is somewhat speculative.
We are agree. The reviewer doesn’t request corrections
L90-92 Ecological competition does not necessarily imply the secretion of growth inhibitors. It could occur simply as a consequence of competition for organic carbon.
We are agree. In the manuscript we enumerate, according to the bibliography, some of the processes, but not all. The reviewer doesn’t request corrections.
L133 Were the new soil samples tested for [Hg]? Metal concentrations in soils can be very heterogenous. If you assume a concentration based on previous measures the concentration should be reconfirmed on your samples.
The new soil samples were no tested in the present study because the plots sampled are characterized research plots. But we agree with you, and we are going to take your advice in account for future studies.
L182-184 By combining all of these samples you have reduced your replication to just three technical reps.
Yes. The Illumina analysis is outsourced. We send to the company the three technical replicas (about 20g of soil for each replica).
L152 consortia?
Corrected.
L168. How much soil was used in each tray?
Each alveoli have a volume of 300 cm3, which each approximately 700 g of soil. This means that each tray contains about 8.5 Kg of soil.
L185 This is not clear. Should be recast.
Changed by: “For each treatment, the rhizospheric soil was gathered (1-2 g per plant) and homogenized to constitute the 60 g analysis sample. The homogenized was divided into three technical replicas”
L204-205 Were the sequences checked for chimeras? What database and version were used for alignments and taxonomic ID? How many OTUs were identified for each sample? How many sequences for each sample? were the samples rarefied?
For the chimera checking were use DADA2 software. The software used for the alignment and taxonomic ID was QIIME2.
The OTUs obtained for each sample were 1,125 and 998 for Cont_S and Cont_P respectively.
We have obtained 166,616,057 raw reads for Cont_S and 115,535,404 raw reads for Cont_P.
The samples were not rarefied.
L187-188 This is incomplete. What happens after centrifugation? Which fraction is then of interest? Supernatant? Pellet? Complete the M&M.
Information added in lines 189 - 190: “The supernatant was collected for the cenoantibiograma study”
L228-233 Repetition.
Deleted.
L227 This is not clear. In the M&M it sounds as though you have metagenomic samples from 10 treatments, not counting the soil control. Yet in this section you present only a soil control and soil rhizosphere.
The metagenomic study was carried out only for the soil control and rhizosphere soil. Information added in lines 189 – 191: “The remaining rhizospheric fraction of non-inoculated plants (approx. 60 g per treatment) and bulk soil (control soil) was separated into three technical replicas for metagenomic study”
It also inidicates in the M&M that there are 3 technical reps but there appears to be no evidence of this in the results.
On line 239, we indicate that we perform a Pearson’s correlation test to evaluate the similarity of the technical replicas obtaining an r > 0.99, which means that the three replicates are clonal.
L210 How was it sown? How much was plated? What do the stats look like on the replicates strips?
Kirby-Bauer method. The results for each strip were homogeneous on each replica.
L240-241 What did the three replicates look like. Not much difference in the Simpson. To prove to the reader that there is a real difference here you need stats on replicates. One can easily see 2-3 fold differences in OTU size just through the variability of PCR amplification.
Fig 1. This figure needs error bars.
We don’t add error bars because the samples are clonal between them.
L263 There is no indication of replication in this experiment. Need standard deviation or comparable. How many reps were there? How many technical reps and how many biological reps?
Of course, this experiment has three technical replicates. But there weren’t any differences between replicas because each strain comes from a plate in pure culture. In the same way, each antibiotic tested comes from the same box, so there is not deviation in the experiment.
L286 Who? Are the samples a Who? Perhaps which would be better.
That’s a grammatical error. Corrected with word: “which”.
Fig 2 legend. More information is needed in the legends. What data set is this PCA based on?)
The data set which this figure is based on, is the MIC obtained from the cenoantibiogram. The load factors (the MIC distribution on the PCA) which determine the distribution of the strain behavior are represented on the Figure 3.
L317 What is this gender? Do you mean genus?
Your are right, it’s a misspelling. The correct word is genus.
L334-335 Somewhat awkwardly expressed
“Changed to: The presence of this phylum has been traditionally associated to aquatic environments but is a singularity in edaphic ecosystems. Its presence is probably related to greater resistance to Hg as other Gram-negative groups”
L397 This seems like a large extrapolation.
Agree but it’s the results obtained.
L442 Needs to be rephrased.
Changed the phrase “The addition of the A1 strain (Brevibacterium frigoritolerans) both isolated and consor-tium (CS1, CS2 and CS3)…” by “The addition of the A1 strain (Brevibacterium frigoritolerans) in isolation and as a consortium (CS1, CS2 and CS3)
Round 2
Reviewer 1 Report
Dear Authors,
From your comments its clear that your most of the work is published previously (mainly PGPB traits, phytoprotection data) . Current study is based mainly on metagenomic analyses where data files are missing.
As you mentioned 'At the moment we are developing some new experiments to analyse the Hg concentration in soils'. Probably would be better if you can complement the data with these results and make a good manuscript to be presented in good journal.
Best Regards
Author Response
From your comments it’s clear that your most of the work is published previously (mainly PGPB traits, phytoprotection data). Current study is based mainly on metagenomic analyses where data files are missing.
First of all, we want to thank the reviewer for the detailed analysis of the work and his constructive and accurate observations. Next, we proceed to try to answer them:
The current work is a totally original, based on the sequencing of amplicons of the edaphic community and the study of the expression of the mechanisms of resistance to antibiotics. The purpose of this paper is to know the impact of the introduction of a new exogenous PGPB species. For this we compare the results (before and after the inoculum) of the aforementioned techniques.
It is true that, throughout the work (especially in the introduction and discussion section), previous studies are mentioned as part of the group's line of research. However, the results of this work have not been published before.
In response to the reviewer's correct observation, the following amendments have been made to the document:
- Elimination of those metagenomic data, which due to not having the Fastaq (due to the closure of the entity that carried out the sequencing and that, although it sent us the results never provided us with these files), the reference numbers could not be provided.
- Modification of the content of the article in order to adapt to the Fastaq files that we have. For more detail, the modifications made have been the following:
- Removed references to Fastaq files, respecting the results of files with reference number in BioPorject PRJNA934906 y PRJNA934908. Having provided the required documents accessible in free format in the links:
- PRJNA934906: https://www.ncbi.nlm.nih.gov/bioproject/?term=PRJNA934906
- PRJNA934908: https://www.ncbi.nlm.nih.gov/bioproject/?term=PRJNA934908
- A more detailed description of the metagenomic analysis of the above files is provided.
- Removed references to Fastaq files, respecting the results of files with reference number in BioPorject PRJNA934906 y PRJNA934908. Having provided the required documents accessible in free format in the links:
- We have proceeded to update the title, whose content is in line with the new wording of the document.
- We have proceeded to the inclusion of a new author whose function has been to new review the metagenomic analysis.
As you mentioned 'At the moment we are developing some new experiments to analyse the Hg concentration in soils'. Probably would be better if you can complement the data with these results and make a good manuscript to be presented in good journal.
The observation of the reviewer is correct and indeed at this moment we are designing new experiments for the determination of Hg in soil, which allow to verify the ability to reduce this contaminant by the action of inoculated bacteria. However, these are experiments that are still underway and we believe that these results will correspond, after analysis, to another new publication.